# Putative Biomarkers of Environmental Enteric Disease Fail to Correlate in a Cross-Sectional Study in Two Study Sites in Sub-Saharan Africa

**DOI:** 10.3390/nu14163312

**Published:** 2022-08-12

**Authors:** Pascale Vonaesch, Munir Winkel, Nathalie Kapel, Alison Nestoret, Laurence Barbot-Trystram, Clément Pontoizeau, Robert Barouki, Maheninasy Rakotondrainipiana, Kaleb Kandou, Zo Andriamanantena, Lova Andrianonimiadana, Azimdine Habib, Andre Rodriguez-Pozo, Milena Hasan, Inès Vigan-Womas, Jean-Marc Collard, Jean-Chrysostome Gody, Serge Djorie, Philippe J. Sansonetti, Rindra Vatosoa Randremanana

**Affiliations:** 1Unité de Pathogénie Microbienne, Institut Pasteur, 25-28 Rue du Dr Roux, 75015 Paris, France; 2Department of Fundamental Microbiology, University of Lausanne, Campus UNIL-Sorge, 1015 Lausanne, Switzerland; 3Human and Animal Health Unit, Swiss Tropical and Public Health Institute & University of Basel, Kreuzstrasse 2, 4123 Allschwil, Switzerland; 4Service de Coprologie Fonctionnelle, Hôpital Pitié-Salpétrière, Assistance Publique-Hôpitaux de Paris, 47-83 boulevard de l’Hôpital, 75013 Paris, France; 5Laboratoire de biochimie métabolique, Hôpital Necker Enfants Malades, Assistance Publique-Hôpitaux de Paris, 149 Rue de Sèvres, 75015 Paris, France; 6Unité d’Epidémiologie et de Recherche Clinique, Institut Pasteur de Madagascar, BP 1274, Antananarivo 101, Madagascar; 7Unité d’Epidémiologie, Institut Pasteur de Bangui, Bangui BP 923, Central African Republic; 8Unité d’Immunologie des Maladies Infectieuses, Institut Pasteur de Madagascar, BP 1274, Antananarivo 101, Madagascar; 9Unité de Bactériologie Expérimentale, Institut Pasteur de Madagascar, BP 1274, Antananarivo 101, Madagascar; 10Cytometry and Biomarkers Unit of Technology and Service, Institut Pasteur and Université Paris Cité, 25-28 Rue du Dr Roux, 75015 Paris, France; 11Complexe Hospitalo-Universitaire de Bangui, Bangui BP 923, Central African Republic

**Keywords:** environmental enteric dysfunction, stunted child growth, Sub-Saharan Africa, anemia, biomarker, alpha-1-antitrypsin, citrulline, calprotectin, lactulose-mannitol test, insulin-like growth factor

## Abstract

Environmental enteric dysfunction (EED) is an elusive, inflammatory syndrome of the small intestine thought to be associated with enterocyte loss and gut leakiness and lead to stunted child growth. To date, the gold standard for diagnosis is small intestine biopsy followed by histology. Several putative biomarkers for EED have been proposed and are widely used in the field. Here, we assessed in a cross-sectional study of children aged 2–5 years for a large set of biomarkers including markers of protein exudation (duodenal and fecal alpha-1-antitrypsin (AAT)), inflammation (duodenal and fecal calprotectin, duodenal, fecal and blood immunoglobulins, blood cytokines, C-reactive protein (CRP)), gut permeability (endocab, lactulose-mannitol ratio), enterocyte mass (citrulline) and general nutritional status (branched-chain amino acids (BCAA), insulin-like growth factor) in a group of 804 children in two Sub-Saharan countries. We correlated these markers with each other and with anemia in stunted and non-stunted children. AAT and calprotectin, CRP and citrulline and citrulline and BCAA correlated with each other. Furthermore, BCAA, citrulline, ferritin, fecal calprotectin and CRP levels were correlated with hemoglobin levels. Our results show that while several of the biomarkers are associated with anemia, there is little correlation between the different biomarkers. Better biomarkers and a better definition of EED are thus urgently needed.

## 1. Introduction

Environmental enteric dysfunction (EED) is an inflammatory syndrome of the small intestine suspected to contribute to stunted child growth and acute undernutrition. This syndrome is characterized through shortening of the intestinal villi, influx of inflammatory cells, especially T-cells into the mucosa, as well as an increase in cell permeability, leading to bacterial translocation [1]. The syndrome is well-known and was first described in travelers spending longer time in the tropics [2,3]. In the last 10 years, EED has become more prominent in the search for more efficient treatments of childhood undernutrition. Thus, there has been an increased effort to identify biomarkers able to discriminate and quantify the presence and severity of EED. The most widely accepted pathway to disease hypothesizes that recurrent symptomatic or asymptomatic pathogen encounters weaken the gut barrier, leading to translocation of bacteria, chronic inflammation, and eventually gut atrophy, gut leakiness and systemic translocation of bacteria [4]. The second hypothesis suggests that inherent changes to the microbiota might be inducing EED [5,6]. 

Currently, the gold standard test to diagnose EED is a biopsy followed by a histological analysis of the samples. In most circumstances, taking biopsies is impractical and invasive, so other ways of detecting EED are needed. Several studies considered biomarkers of EED and correlated them with growth measures [7,8,9,10,11,12,13], such as height-for-age z-score or biopsy-diagnosed EED [6,14]. Only a few studies compared different biomarkers with each other [6,14]. Indeed, a systematic meta-analysis in 2018 suggested that there is little correlation between the different domains suggested to jointly define EED [15], though this has never been extensively studied in a single study. There are also conflicting data on if and how these biomarkers are influenced by micronutrient deficiencies, such as anemia [14,16,17,18], that are frequently observed in undernourished children or how, on the other hand, EED might influence and/or aggravate micronutrient deficiencies. This is important for both interpreting the biomarkers measured and assessing if anemia and EED are independently associated with stunting and need to be treated jointly for a maximum success of treatment outcome.

This project aimed to compare putative biomarkers of EED in two groups of children included in a cross-sectional study: those with and without stunted growth. Furthermore, the project aimed to assess if these putative biomarkers are correlated with each other and if the associations are modulated by anemia.

We show that there is little correlation between the different biomarkers in either of the two study locations, that most of the biomarkers do not correlate with stunted child growth but that anemia is independently associated with several putative biomarkers of EED.

## 2. Materials and Methods

### 2.1. Study Setup and Sample Collection 

This project was part of the Afribiota project, a cross-sectional study on stunting performed in children aged 2–5 years in Antananarivo, Madagascar and Bangui, Central African Republic. All children with valid biomarker data were included in this study. The subjects included are summarized in the flowchart in Figure 1A. The detailed study protocol [19] as well as the associated metadata on social and environmental factors [20] and enteropathogen carriage [21] have been reported previously. In brief, children living in Bangui, Central African Republic or Antananarivo, Madagascar, aged 2–5 years and not experiencing any severe disease (acute respiratory distress, HIV, watery or bloody diarrhea, acute undernutrition) were included. In each country, the target size was 460 children, classified as severely stunted, moderately stunted and not stunted according to the median height of the WHO reference population [22]. Metadata included nutritional status, age, socio-economic factors, ferritin and hemoglobin levels as well as parasite and enteropathogen load, and these were assessed using a standardized questionnaire and routine diagnostics. For all subjects, we collected blood and feces and for undernourished subjects, we collected duodenal and gastric aspirates. Biobanking was performed by the Clinical Investigation and Access to BioResources Platform (ICAReB) at the Pasteur Institute, Paris, and by the Pasteur Institutes of Madagascar and Bangui. 

### 2.2. Measurement of Biomarkers in Feces, Duodenum and Blood

#### 2.2.1. Blood Biomarkers

Complete blood count, C-reactive protein (CRP) and ferritin levels were measured at the Clinical Biology Center of the Institut Pasteur de Madagascar and the Laboratoire d’Analyse Médicale at the Institut Pasteur de Bangui according to accredited methods. Ferritin levels were corrected for systemic inflammation based on CRP values [23]. Hemoglobin values of Malagasy children were adjusted for altitude as described in [24,25]. Citrulline and other amino acids were measured with liquid chromatography coupled to tandem mass spectrometry (UPLC-MS/MS). For accurate quantification, stable isotope internal standards of the same structure for each amino acid (purchased from Eurisotop, Saint Aubin, France) were added to the samples before protein precipitation. Samples were first derivatized using the AccQ Tag^TM^ Ultra (Waters Corporation, Milford, MA, USA) according to manufacturer recommendations. Amino acid separation was performed with an Acquity™ UPLC system using a CORTECS™ UPLC C18 column (1.6 µm, 2.1 × 150 mm) coupled to microTQS™ tandem mass spectrometer (Waters Corporation, Milford, MA, USA).

Endocab levels were measured on a subset of samples in Madagascar. To this purpose, plasma samples were tested for anti-LPS IgM, IgA and IgG using a commercially available ELISA kit (Endocab Human, Hycult Biotechnology, Noord-Brabant, The Netherlands). Plasma samples were used undiluted for IgM and IgA and diluted at a ratio of 1:400 and 1:200 for measuring IgG. The assay is based on a solid-phase sandwich ELISA to detect and quantify antibodies against LPS (endotoxin). A standard curve was created for each plate using standards provided by the manufacturer. 

Immunoglobulin levels were measured using commercial LUMINEX assays (Biorad, Bio-PlexPro™ Human Isotyping Panel, 6-plex, ref. 171A3100M for IgG1-4, IgM, IgE and Bio-Plex Pro™ Human IgA Isotyping Assay, ref. 171A3101M, Herkules, CA, USA). The assays were measured on a MAGPIX system (LUMINEX, Ghent, Belgium) according to manufacturer instructions. Values were measured in duplicates, and all samples with more than 20% difference in between the two measurements or values below or above the standard curve were repeated. Plasma samples were diluted 1/40,000 (IgG 1-4, IgA, IgM) and 1/500 (IgE). 

Cytokine levels were measured on a subset of samples from Madagascar according to manufacturer instructions using the Cytokine 30-Plex Human Panel from Invitrogen (ref. LHC6003M, Thermofisher Scientific, Waltham, MA, USA) using plasma samples diluted 1/2 prior to measurement and the same LUMINEX setup as described above.

Igf1 levels were measured at Eurofins Biomnis using a chemiluminescence assay on samples from Bangui only. 

#### 2.2.2. Duodenal and Fecal Biomarkers

Of note, duodenal samples were available only for stunted children. To assess for small intestine bacterial overgrowth (SIBO), duodenal samples with a pH of at least 5 were cultured aerobically and anaerobically in different rich media as described previously [5]. Cultures were considered positive for SIBO if the total bacterial count exceeded 2 × 10^5^ CFU per mL of duodenal fluid [26]. Fecal and duodenal calprotectin and alpha-1-antitrypsin (AAT) concentrations were assayed at the coprology laboratory of the Pitié Salpêtrière Hospital according to standard accredited procedures as described previously [27]. Duodenal data for AAT and calprotectin was retrospectively excluded from the analysis due to a high number of missing data.

Immunoglobulin levels were measured using commercial LUMINEX assays (Biorad, Bio-PlexPro™ Human Isotyping Panel, 6-plex, ref. 171A3100M for IgG1-4, IgM, IgE and Bio-Plex Pro™ Human IgA Isotyping Assay, ref. 171A3101M, Herkules, CA, USA). The assays were measured on a Bioplex 200 system in combination with the DropArray approach (Curiox Biosystems Pte Ltd., Singapore, Singapore) according to manufacturer instructions. Values were measured in duplicates, and all samples with more than 20% difference in between the two measurements or values below or above the standard curve were repeated. Final values were normalized to the initial fecal weight used for extraction. Analysis was performed with Bio-Plex Manager Software version 6.1.1 (Biorad, Herkules, CA, USA). Duodenal samples were used undiluted (IgG1-4, IgM) and at a dilution of 1/100 in an extraction cocktail (IgA). For feces, 200 mg of fecal samples were homogenized in 1 mL of PBS containing Protease Inhibitor Cocktail (Roche Diagnostics GmbH, Mannheim, Germany), incubated for 30 min on ice and centrifuged for 10 min at 10,000 g at 4 °C. The supernatant was then used in downstream applications at a dilution of 1/100–1/10,000. 

Cytokine levels in the feces were measured using the Cytokine 30-Plex Human Panel from Invitrogen on the same, undiluted protein extracts using the same LUMINEX setup as described above. Overall fecal cytokine values were, however, very low and thus not included in the final analysis.

Bacterial pathogen load was determined using quantitative PCR on a set of bacterial virulence genes (*ompC, ipaH, estla, eltB, eae, bfpA, aggR, aaiC, cadF, ctxA*) as published previously [21]. Presence of a given gene was based on a Ct value < 37, and the total number of genes present in each fecal sample was reported. Parasites were identified using microscopy as well as qPCR-based approaches as described in [28]. 

#### 2.2.3. Urine Biomarkers

Lactitol–mannitol levels in urine were assessed as described in [29]. Briefly, the dual sugar absorption test was performed in the morning following an overnight fast and after discarding all overnight urine. The children were then given a mixture of 10% mannitol and lactitol in mineral water at a dosage of 0.1 g/kg of body weight for each sugar. The solution was given to them within 30 min, and the children were given mineral water after a 2 h wait. All urine was collected for the 5 following hours with gentamicin as a preservative. The final urine volume was measured, and the urine mixed as an aliquot was stored at −80 °C unitl analysis. The samples were analyzed at Hôpital Pitié Salpetrière using a Perkin Elmer AutoSystem KL (Perkin Elmer, Waltham, MA, USA) gas chromatograph equipped with a flame ionization detector. Samples were pre-treated as follows: 3 mL of urine was deionized using 0.5 g of Amberlite IRN-150 resin, then centrifuged for 10 min at 1500× *g*. Of this deionized urine, 950 ul were mixed with 50 μL of internal standard [a mixture of inositol and turanose (*v*/*v*) at a final concentration of 5 mg/mL], lyophilized and submitted to derivatization by addition of 400 μL of a mixture composed of hexamethyldisilazane and trimethylchlorosilane in pyridine (3:1:9) (Supelco analytical, Bellefonte, PA, USA) for 1 h at 60 °C. Samples were extracted using 200 μL hexane and injected in a capillary column (30 m Å~0.25 mm ID Elite N 9316076, Perkin Elmer) with helium as gas carrier at a 1.5 bar head pressure. The temperature program was performed from 150 to 250 ÅãC with 20 ÅãC per min. Peak areas were automatically integrated, and mannitol and lactitol areas reported to inositol and turanose areas, respectively. Results were expressed as mg/L and IP (%) as L/M ratio (L/M, %). L/M ratio data was retrospectively excluded from the analysis due to a high number of missing data (Appendix A).

#### 2.2.4. Cutoffs Used for the Categorization of the Biomarkers

We used the following cutoffs to transform part of the biomarkers into categorical variables: anemia was defined as less than 110 g of Hb/l of blood according to the WHO criteria [30,31]. For AAT, values below 1.25 mg/g of fecal dry weight or below 0.15 of fecal wet weight were considered normal. For calprotectin, thresholds were adapted to age and were considered normal if the calprotectin levels were below 150 mg/g of fecal wet weight for children aged 2–3 years and 100 mg/g of fecal wet weights for children aged three years and older. These values are based on the normal values used at the Hôpital Pitié Salpêtrière. Low ferritin levels were defined as <12 μg/L in the absence of inflammation [32]. To eliminate the influence of inflammation on ferritin plasma concentrations, a correction factor of 0.67 was used to adjust its values in the presence of inflammation [23]. Citrulline values were defined according to the normal values provided by the Hôpital Necker Enfants Malades as follows: low citrulline: <7 umol/L, elevated citrulline: >43 umol/L. Amino acids were categorized according to the normal values provided by the Hôpital Necker Enfants Malades as follows: low alanine: 151 umol/L, elevated alanine: >407 umol/L; low valine: <133 umol/L, elevated valine: >349 umol/L; low isoleucine: <37 umol/L, elevated isoleucine: >85 umol/L; low leucine: <67 umol/L, elevated leucine: >175 umol/L. High CRP levels were defined as: >6 mg/L. 

### 2.3. Statistical Analysis

The metadata can be found in Appendix A. Data analysis was performed in R, version 4.1.2. Missing variables were reported and putative biomarkers with high levels of missing values were excluded from the models (i.e., cytokines in the blood, lactitol–mannitol and small intestine inflammation measurements). The statistical analyses for the different sections are indicated below, and the analysis code can be found on Github under VonaeschLabUNIL/Afribiota (https://github.com/VonaeschLabUNIL/Afribiota.git; accessed on 5 July 2022). All variables were tested for interactions, and interaction terms were included in the final multivariate models. 

#### 2.3.1. Correlation of Biomarkers across Body Sites and Between Each Other

Binary comparisons were made using the Student’s t-test or Spearman correlation for continuous variables and the Chi Squared Test for proportions and illustrated using correlation plots, heatmaps or boxplots. Furthermore, we corrected *p*-values for multiple comparisons using the False Discovery rate (FDR). When appropriate, a square-root or logarithmic transformations of the response variables were taken so that the residual terms were closer to being normally distributed. Multivariate comparisons were illustrated using Prinicipal Component Analysis. 

#### 2.3.2. Factors Associated with Anemia and Height-for-Age Z-Score

Binary comparisons between the biomarkers and anemia or height-for-age z-score were made using the Student’s t-test or Spearman correlation for continuous variables and the Chi Squared Test for proportions, and *p*-values were corrected by the FDR for multiple comparisons. When appropriate, a square-root or logarithmic transformations of the response variables were taken so that the residual terms were closer to being normally distributed. Furthermore, for computing significance in correlations between two variables/ biomarkers, we used the Spearman rank correlation method, which is a nonparametric measure of rank correlation that is ideal for variables that are skewed or have a non-normal distribution. Last, height-for-age z-score was modeled as a linear function of variables of interest accounting for a priori controls including country of origin, sex, age, food diversity score and season of inclusion. To avoid a disproportionate effect of potential outliers in the dataset (Appendix A), which might have a large influence on the parameter estimates from the linear models and assess for robustness of the associations, we applied a bootstrapping approach.

#### 2.3.3. Factors Associated with Putative Biomarkers of EED

Continuous response variables were modeled as a linear function of variables of interest accounting for a priori controls including a bootstrapping approach.

Linear models were corrected for the following potential confounding variables: country of inclusion, season of inclusion, food diversity score, age, gender and, for some of the models, hemoglobin levels. Inflammation-related biomarkers were also corrected for the number of enteropathogenic virulence genes detected as well as carriage of *Trichuris trichura* and/or *Ascaris lumbricoides*.

## 3. Results

### 3.1. Description of Study Population and Biomarkers Measured

A total of 804 subjects were included in this analysis; 417 were from Antananarivo (215 non-stunted and 202 stunted) and 387 from Bangui (216 non-stunted and 171 stunted) (Figure 1A). Duodenal biomarkers were available only for stunted children. Furthermore, given biomarkers including Igf1 (Bangui, Central African Republic) as well as blood cytokines and endocab levels (Antananarivo, Madagascar) were performed only on two small subsets of participants. The study population and relevant metadata are described in Table 1. 

Several putative biomarkers of EED were measured within this study: citrulline reflecting enterocyte mass; alpha-1-antitrypsin, reflecting protein-losing enteropathy; calprotectin, reflecting intestinal inflammation; immunoglobulin levels in the duodenum, the feces and the blood as well as cytokine levels in the blood as general markers of inflammation; the lactitol–mannitol ratio, reflecting intestinal permeability; endocab levels, reflecting translocation of Gram-negative bacteria and Igf1 (insulin-like growth factor) and branched-chain amino acids as general read-outs for undernutrition. 

Igf1 values were analysed in a sub-study in Bangui (n = 364 children), blood cytokine values in a sub-study in Madagascar (n = 66) and endocab values in a subset in Madagascar (n = 61) (Appendix A). The lactitol–mannitol ratio was a posteriori excluded from the analysis due to very high missing or uninterpretable data. The values of the putative biomarkers between the study countries are summarized in Table 2 and Appendix A and the sample size numbers for each biomarker in Figure 1B. 

#### 3.1.1. Enterocyte Mass and (Intestinal) Inflammation

A total of 3.2% (13/410) in Antananarivo and 8.3% (32/384) children in Bangui, respectively, showed citrulline levels below the normal value. A total of 6.4% (27/422) of children in Antananarivo and 2.1% (8/387) of children in Bangui, respectively, had elevated levels for alpha-1-antitrypsin (AAT), and 36.4% (144/396) of children in Antananarivo and 24.4% (87/357) of children in Bangui had elevated levels of calprotectin.

Immunoglobulin levels were generally low in the duodenum with higher levels of IgG1 and IgG2 in the feces and blood (Appendix A). Cytokine levels were measured on a subset of children from Madagascar (n = 66). The profile of blood cytokines did not differ between children of differing nutritional states (Appendix A). Immunoglobulin levels were measured in the blood, duodenum and feces in both study sites. Overall, the immunoglobulin values in samples from the Central African Republic were more diverse than from Madagascar, yet there was no visible clustering by stunting status (Appendix A). A total of 8.0% (34/423) and 21.0% (71/345), in Bangui and Antananarivo, respectively, showed elevated levels of CRP.

#### 3.1.2. Gut Leakiness and Bacterial Translocation

The test for the lactitol–mannitol ratio had many missing observations or yielded non-interpretable results since the mannitol levels were below the detection threshold in many children. The remaining values were almost exclusively above the threshold set for normal gut permeability, suggesting that the observed children suffered from leaky gut. Given the large number of missing values, we did not include this variable in our models. There was only limited data available for endocab (n = 61) and only from the Madagascar study site. The substudy comprised both stunted (n = 36) and non-stunted (n = 25) children. We did not observe any significant association between stunted child growth and either of the sub-forms of endocab (IgA (*p* = 0.17), IgG (*p* = 0.42), IgM (*p* = 0.30).

#### 3.1.3. General Markers of Undernutrition

Igf1 levels were measured in Bangui only (n = 367) and did not differ in between the stunted and non-stunted children (*p* = 0.35). Branched chain amino acids were significantly lower in stunted compared to non-stunted children (*p* < 0.001 on the geometric mean of the branched chain amino acids).

Overall, in our study group, while roughly one third of all children showed signs of general gut inflammation (calprotectin), only very few children showed signs of protein-losing enteropathy (AAT) or a decrease in enterocyte mass as assessed through citrulline levels.

### 3.2. Little Correlation of Biomarkers across Body Sites and between Each Other

Environmental enteric dysfunction mainly affects the small intestine. We wondered how markers in the blood or the feces, which are more readily available on a routine basis, are correlated with measurements of inflammation in the small intestine. We compared biomarkers of inflammation in the small intestine, feces and blood. 

#### 3.2.1. Correlation across Body Sites

Data on immunoglobulins were available for 82 children in the blood, feces and duodenum; 90 children in duodenal and fecal samples and 594 children in blood and fecal samples. As expected, a principal components analysis (PCA) plot showed that the samples clustered by analysis site, with duodenal and fecal samples showing similar profiles that were distinct from the profiles in the blood (Appendix A). When comparing the levels of immunoglobulins between the different body sites using a two-way comparison (Spearman correlation), after adjusting the *p*-values for multiple comparisons (within a compartment), there was a significant association with IgG4 between blood and fecal samples. However, the association was weak and mostly driven by a few observations. Immunoglobulin A, G and M levels in the blood and immunoglobulin G and M levels in the feces correlated with each other more strongly (Appendix A).

Furthermore, there was no correlation between intestinal inflammatory markers in the small intestine and the feces when correcting for age, sex and country of origin whether using the untransformed values (Figure 2A,B) or transformed values. Endocab levels as well as blood cytokine levels were only measured in a subset of children from Madagascar, and there was thus only a very low number of children with shared cytokine and endocab data in the blood (n = 8), not allowing for statistical comparisons. 

#### 3.2.2. Correlation between Different Biomarkers

EED is defined through several hallmarks, including intestinal inflammation, gut atrophy, and increased gut permeability. We aimed to compare the levels of the different biomarkers for each child.

For 608 children, we had data to compare AAT levels, calprotectin levels, citrulline levels and control for potential confounding factors including age, gender, country of origin, anemia, BCAA levels and systemic inflammation. For 299 children from Bangui, we also had data for Igf1 levels.

In the group of 608 children, as expected, AAT and calprotectin levels, both reflecting different aspects related to intestinal inflammation, were significantly associated between each other (Figure 2C). Furthermore, there was an inverse correlation between citrulline and CRP levels (Figure 2G) and a direct correlation between BCAA levels and citrulline levels (Figure 2F). However, there was no significant association between citrulline levels and AAT and/or calprotectin nor, in the group of 299 children, Igf1 and any of the other putative biomarkers (Figure 3).

Thus, in the children studied, we found no strong associations between inflammation markers in the duodenum, the feces, or the blood nor among the different putative biomarkers of EED.

### 3.3. Blood Amino Acid Levels, Blood IgE and Ferritine Are Associated with Stunted Child Growth

EED has been postulated to be associated with stunted child growth. We tested the putative EED markers for a possible association with the HAZ-score in continuous or categorical form. For blood amino acid levels in a bivariate analysis, the only variables associated with stunted child growth were: citrulline (categorical: *p* = 0.034; continuous: *p* < 0.001, fold change stunted/non-stunted: 0.93), valine (categorical: *p* = 3.6 × 10^−5^, continuous: *p* = 1.1 × 10^−6^); leucine (categorical: *p* = 1.8 × 10^−4^, continuous: *p*= 1.1 × 10^−6^), isoleucine (*p* = 2.8 × 10^−7^, continuous: *p* = 7.9 × 10^−5^), alanine (*p* = 0.008, continuous: *p* = 0.9), BCAA (continuous: *p* = 1.4 × 10^−7^, fold change stunted/non-stunted: 0.91) and CRP as a categorical variable (*p* = 0.006). Using linear models correcting for age, gender and country of origin, there was no association between fecal alpha-1-antitrypsin levels, fecal calprotectin levels, CRP, Igf1 or endocab and HAZ-score (Figure 4A,C,D). Testing for associations between the different immune variables (blood cytokines, immunoglobulins), IgE, VEGF and Il2r levels in the blood had unadjusted *p*-values of less than 0.05; however, except for IgE (*p* = 0.03), associations with the HAZ-score were not significant in linear models after adjusting for additional factors such as country of origin, stunting status, age, gender, season of inclusion and clinical factors (clogged or runny nose, cough, hemoglobin) (Appendix A).

There was a significant linear association between citrulline levels and HAZ-score (Figure 4A) as well as BCAA levels and HAZ-score (Figure 4D). This association was, however, confounded, as significance was lost when the models included levels for both citrulline and BCAA levels. In this case, only BCAA remained associated with stunting, suggesting the initial finding is due to the interaction between citrulline and BCAA levels (Figure 4C,D).

Thus, from the biomarkers tested, only plasma amino acids, ferritin and blood IgE levels were associated with stunted growth in our study group.

### 3.4. Factors Associated with Putative Biomarkers of EED

We next assessed for factors associated with the putative biomarkers measured (Figure 3). 

We found citrulline levels to be significantly and negatively associated by CRP levels and positively associated with hemoglobin levels and BCAA levels. BCAA levels were associated with food diversity score, country of inclusion, CRP and citrulline levels.

Furthermore, we found AAT levels to be associated with age, hemoglobin, and calprotectin levels. Calprotectin was in turn significantly associated with AAT levels, season of inclusion of the child, the number of enteropathogen virulence genes detected as well as hemoglobin levels.

IgE levels in the blood were associated with *Trichuris* infection, gender, and age. Igf1 levels were associated with the age of the child and CRP levels with season of inclusion (Figure 3). Finally, ferritin levels were associated with age, country of inclusion, calprotectin, hemoglobin, and CRP.

Indeed, all the main biomarkers measured in our study except BCAA and Igf1 were directly correlated with hemoglobin levels (Figure 5).

Thus, our data reveals that the putative EED markers of intestinal inflammation and enterocyte mass are not correlated with each other but are heavily influenced by comorbidities such as anemia and environmental factors such as season of inclusion, age and diet.

## 4. Discussion

In the group of children analyzed, we found AAT to correlate with calprotectin levels, CRP with citrulline levels and citrulline levels with BCAA levels. Furthermore, BCAA, citrulline, ferritin, fecal calprotectin and CRP levels were correlated with hemoglobin levels. We also saw correlation between BCAA levels, blood IgE values and ferritin and stunted child growth. Our results show that while several of the biomarkers are associated with anemia, there is little correlation between the different biomarkers. A better definition of EED as well as more specific biomarkers are thus urgently needed.

The results presented in this article in light of the accumulated evidence raise an important question: is EED really a single etiological entity or is it an accumulation of different pathophysiological changes affecting the gut that are context dependent? Our results suggest the different biomarkers referring to different domains of EED are not related, a result that is confirmed on a smaller subset of EED markers in a systematic meta-analysis by Harper et al. [15]. Indeed, the different hallmarks of EED seem to be independent of each other and heavily influenced by environmental factors such as (asymptomatic) pathogen carriage, micronutrient deficiencies or diet and environmental factors, thus making it difficult to diagnose and even define EED across different contexts with the currently available, non-invasive biomarkers.

We further show that several of the putative biomarkers including BCAA, citrulline, ferritin, fecal calprotectin and CRP levels are associated with hemoglobin as well as calprotectin and CRP with ferritin. A previous cross-sectional study in Bangladesh on children aged two years shows a negative correlation between AAT levels and ferritin [17], and another cross-sectional study in Uganda on children aged six months shows association between higher anti-flagellin IgA, anti-flagellin IgG and anti-LPS IgA concentrations and lower hemoglobin levels [33]. These results jointly point towards a role of EED in micronutrient deficiency or—on the inverse or in addition—a role of micronutrient deficiency on the development or aggravation of EED. Interestingly, the most common extraintestinal complication of intestinal bowel disease (IBD), a syndrome that shares a lot of similarities with EED, is anemia [34]. Further research on how EED is modulated and potentially aggravating micronutrient status is warranted, especially in longitudinal studies in children of the most vulnerable age between conception to the second year of life. 

In our study, while inflammation was not associated with stunted child growth, several of the inflammation markers differed by season and by country of inclusion of the children. This might be due to the high burden of enteropathogens overall in the study area [5,21] as well as the overall pathogen exposure, as shown by the correlation between the number of asymptomatic enteropathogens present and fecal markers of inflammation. As stunting is a long-term syndrome, previous infection rather than current infections might be more prone to be associated with stunted child growth. Associations between enteropathogens and fecal inflammation have been shown in other settings [35,36]. In several previous studies, inflammation markers were also associated with stunted child growth [6,12,13,14,15,37,38], while no association was observed with biopsy-confirmed EED [6]. While we did not observe an association between parasite carriage and stunting in the group of children included in Madagascar [28], there was a clear association of blood IgE levels with stunting, suggesting that previous contact with parasites or allergens is associated with stunted child growth. Thus, it is plausible that inflammation markers are only associated with stunted child growth if infection with enteropathogens is indeed the (or at least one of the) driving forces underlying chronic undernutrition and stunted child growth. Furthermore, previous infection might have an effect on current stunted growth even if the actual inflammation has resolved. This is especially plausible as our study group was quite old, 2–5 years, while the bulk of growth delay normally occurs prior to this age [39]. Surprisingly, we did not detect any association between IgA levels and stunted child growth. While this is counter-intuitive, it is in line with a previous article from the group showing that IgA-coating of bacteria is not significantly changed between stunted and non-stunted children [27]. Our study did not find any correlation between AAT and calprotectin levels in the small intestine and in the feces. In a publication by Chen et al., the authors also did not find any association between fecal immune markers and histology-confirmed EED. As EED is a small intestine disease, these results question the usefulness of fecal markers of inflammation and enteropathy as a readout of EED.

We find an association between citrulline levels and HAZ. This correlation is, however, heavily confounded by the overall protein intake, reflected through the levels of BCAA. This calls for a careful evaluation of citrulline as a biomarker, as citrulline levels alone are not able to distinguish between general problems of undernutrition and actual enterocyte mass loss induced by inflammatory syndromes such as enteropathies. Future longitudinal studies should include more markers of general undernutrition as well as different blood markers, assessing for enterocyte mass.

One of the current most widely used tests, the dual-sugar absorption test, is very difficult to perform in the field, as children need to fast for an entire night and to provide urine samples over several hours [40]. In our study, we encountered the same problems in collecting this data and faced additional issues with measuring mannitol levels, which were very low. Thus, we do not see the dual-sugar absorption test being useful in the field. Overall, previous work offered conflicting data about specific biomarkers associated with stunted child growth [6,7,8,9,10,11,36,40,41,42]. A recent study assessing for biomarkers in a longitudinal cohort of 416 children over two years found little association between classical biomarkers of EED as well as height-for-age score, which is in line with the results we present here. However, there is general consensus that ferritin levels are associated with stunted child growth [9,14,16,17,43], a finding that is replicated in our study. 

Our study has some obvious limitations: first, stunting and EED being gradual and chronic syndromes, a longitudinal design would be more appropriate to infer causal relationships between the biomarkers and stunted growth. The negative results regarding association of biomarkers with stunted child growth might thus simply reflect the fact that pathophysiological changes leading to stunted child growth might have happened in the past and resolved in the meantime. Only a longitudinal approach could clarify this point.

Furthermore, we were unable to perform biopsies, the current gold standard for the diagnosis of EED and also did not manage to obtain reliable results from the lactitol–mannitol test, which is the current reference test to assess for leaky gut. Despite these limitations, our study adds important data on the association between different biomarkers and is one of the cross-sectional studies with the largest breath of biomarkers measured to date.

Overall, based on our results and the recent literature, the most reliable biomarker still seems to be a biopsy followed by histological analysis, where a clear score could be defined in a recent publication across several study sites. However, the histological score also showed site-specific characteristics [44]. More studies in different settings are needed, especially ones that compare the biomarkers at distant body sites to the actual gold standard for EED, small intestine biopsy and histology [44,45,46,47]. Furthermore, studies should include several biomarkers that are then correlated with each other. Finally, studies should have a longitudinal design to infer causal relationships and follow the children over and beyond the most important 1000 first days of life.

## 5. Conclusions

In this cross-sectional study, we show that the putative biomarkers do not correlate with each other and are heavily influenced by confounding factors such as asymptomatic enteropathogenic carriage, age, overall nutritional status and study setting. We further show that BCAA, citrulline, ferritin, fecal calprotectin and CRP levels are correlated with hemoglobin levels and calprotectin and CRP with ferritin. Our data suggest that EED is an assembly of different clinical factors rather than a precise etiological entity, which is not sufficiently explained by current biomarkers. Our study highlights the importance of the study location in the etiology of stunting.

## Figures and Tables

**Figure 1 nutrients-14-03312-f001:**
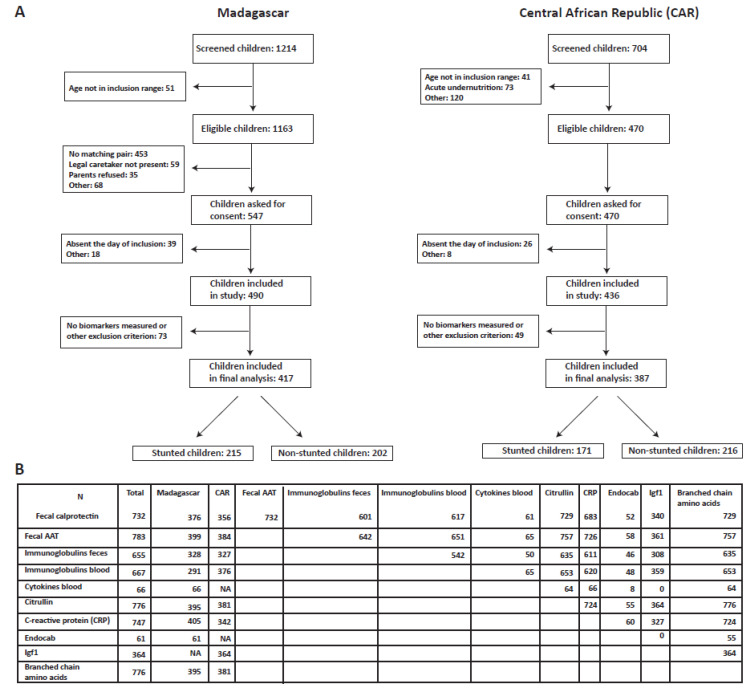
Subjects and samples included in this study: (**A**) flowchart of the subjects included in this study; (**B**) summary of the samples available for each biomarker/ combination of biomarkers.

**Figure 2 nutrients-14-03312-f002:**
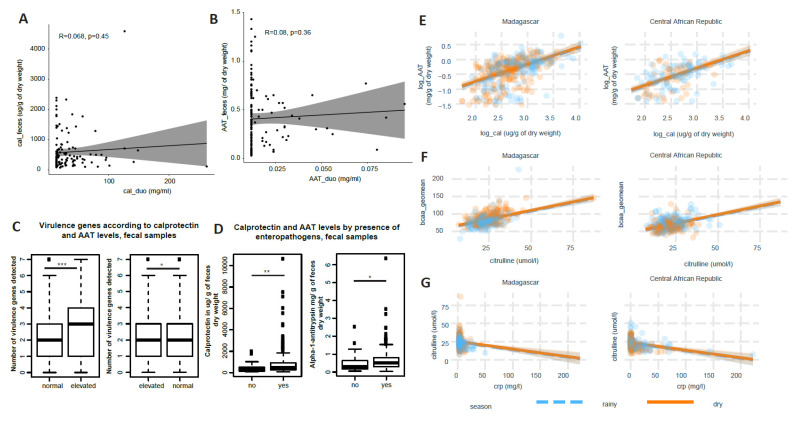
Correlation of different biomarkers between body compartments, with each other as well as with the number of virulence genes: (**A**) correlation plot of calprotectin levels in feces and duodenum; (**B**) correlation plot of alpha-1-antitrypsin (AAT) levels in feces and duodenum; (**C**) number of enteropathogenic virulence genes detected according to calprotectin and AAT levels; (**D**) calprotectin and AAT concentration measured in the feces depending on the presence of at least one virulence gene measured by qPCR; (**E**) correlation of levels of fecal calprotectin and AAT levels; (**F**) correlation of citrulline levels and the geometric mean of branched-chain amino acids (BCAA = isoleucine, leucine, valine); (**G**) correlation of blood citrulline and CRP levels; correlations are based on non-parametric Spearman correlation; *: *p* < 0.05; **: *p* < 0.01; ***: *p* < 0.001.

**Figure 3 nutrients-14-03312-f003:**
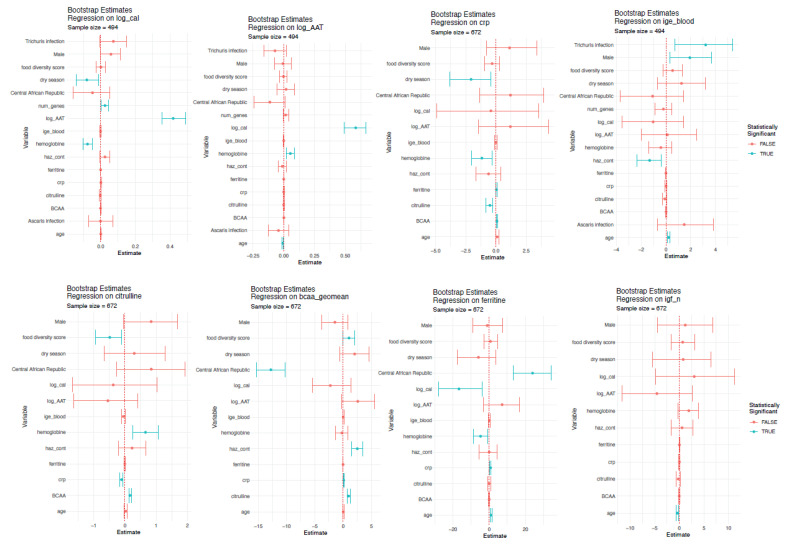
Factors associated with putative biomarkers of EED. Regression models are shown per biomarker, considering associations with other biomarkers as well as potential confounding factors. Significant associations (*p* < 0.05) are indicated in blue.

**Figure 4 nutrients-14-03312-f004:**
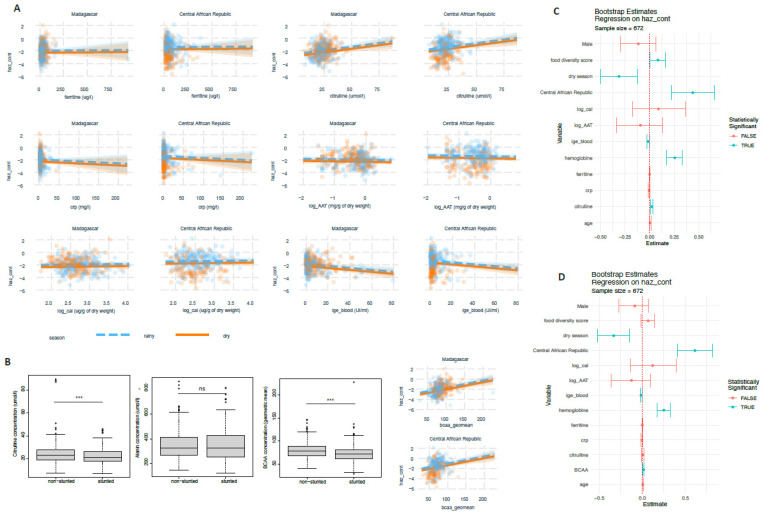
Putative biomarkers and their association with growth delay: (**A**) association of different biomarkers with height-for-age z-score; (**B**) blood amino acid levels as a function of stunting status; (**C**) regression results of the main biomarkers and potential confounders and stunting; (**D**) regression results of the main biomarkers and potential confounders including branched-chain amino acids (BCAA) and stunting. Significant associations (*p* < 0.05) are indicated in blue. Correlations are based on non-parametric Spearman correlation; ***: *p* < 0.01, ns: *p* > 0.05.

**Figure 5 nutrients-14-03312-f005:**
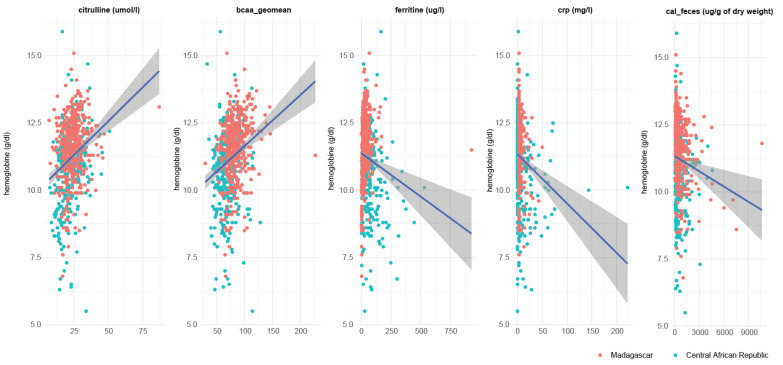
Correlation between putative EED biomarkers and hemoglobin levels.

**Table 1 nutrients-14-03312-t001:** Characteristics of the study population.

	Antananarivo, Madagascar	Bangui, Central African Republic
	(n = 417)	(n = 387)
**Height-for-age z-score**		
**Moderately stunted**	109 (26.1%)	85 (22.0%)
**Mon-stunted**	215 (51.6%)	216 (55.8%)
**Severely stunted**	93 (22.3%)	86 (22.2%)
**Mean (SD)**	−2.03 (1.11)	−1.83 (1.40)
**Median [Min, Max]**	−1.97 [−5.23, 2.03]	−1.75 [−5.67, 2.17]
**Age (months)**		
**Mean (SD)**	42.8 (10.7)	40.0 (10.2)
**Median [Min, Max]**	43.2 [24.2, 60.0]	38.8 [24.2, 60.8]
**Hemoglobin (g/L)**		
**Mean (SD)**	11.6 (1.15)	10.8 (1.42)
**Median [Min, Max]**	11.7 [6.80, 15.1]	11.0 [5.50, 15.9]
**Missing values**	9 (2.2%)	47 (12.1%)
**Anemia (Hb < 11 g/L)**		
**No**	308 (73.9%)	181 (46.8%)
**Yes**	100 (24.0%)	159 (41.1%)
**Missing values**	9 (2.2%)	47 (12.1%)
***Ascaris* infestation**		
**No**	205 (49.2%)	337 (87.1%)
**Yes**	209 (50.1%)	1 (0.3%)
**Missing values**	3 (0.7%)	49 (12.7%)
***Trichuris* infestation**		
**No**	138 (33.1%)	338 (87.3%)
**Yes**	276 (66.2%)	0 (0%)
**Missing values**	3 (0.7%)	49 (12.7%)
***Giardia* infestation**		
**No**	316 (75.8%)	274 (70.8%)
**Yes**	98 (23.5%)	64 (16.5%)
**Missing values**	3 (0.7%)	49 (12.7%)
**Number of virulence genes detected**		
**Mean (SD)**	2.51 (1.52)	0.824 (0.947)
**Median [Min, Max]**	2.00 [0, 7.00]	1.00 [0, 4.00]
**Missing values**	3 (0.7%)	205 (53.0%)
**Food diversity score**		
**Mean (SD)**	3.90 (1.11)	3.63 (1.24)
**Median [Min, Max]**	4.00 [1.00, 7.00]	4.00 [1.00, 7.00]
**Season of inclusion**		
**Rainy season**	199 (47.7%)	228 (58.9%)
**Dry season**	218 (52.3%)	(41.1%)

**Table 2 nutrients-14-03312-t002:** Description of the main putative biomarkers.

	Antananarivo, Madagascar		Bangui, Central African Republic	
	Non-Stunted	Stunted	Non-Stunted	Stunted
	(n = 215)	(n = 202)	(n = 216)	(n = 171)
**Ferritin (ng/mL)**				
Mean (SD)	35.7 (29.6)	35.0 (68.5)	65.5 (74.9)	67.7 (62.7)
Median [Min, Max]	28.0 [1.88, 170]	25.4 [1.27, 929]	40.0 [3.00, 534]	52.0 [4.00, 343]
Missing values	4 (1.9%)	8 (4.0%)	17 (7.9%)	22 (12.9%)
**Ferritin level**				
Missing values	4 (1.9%)	8 (4.0%)	17 (7.9%)	22 (12.9%)
No	175 (81.4%)	158 (78.2%)	189 (87.5%)	138 (80.7%)
Yes	36 (16.7%)	36 (17.8%)	10 (4.6%)	11 (6.4%)
**Citrulline (umol/l)**				
Mean (SD)	25.1 (7.52)	23.2 (6.79)	22.5 (6.52)	21.0 (6.61)
Median [Min, Max]	24.4 [10.5, 87.0]	22.1 [7.00, 46.0]	21.9 [7.50, 50.9]	20.8 [7.55, 44.9]
Missing	7 (3.3%)	15 (7.4%)	2 (0.9%)	4 (2.3%)
**Citrulline level**				
Missing	7 (3.3%)	15 (7.4%)	2 (0.9%)	4 (2.3%)
Normal	200 (93.0%)	182 (90.1%)	200 (92.6%)	152 (88.9%)
Too low	8 (3.7%)	5 (2.5%)	14 (6.5%)	15 (8.8%)
**Valin (umol/L)**				
Mean (SD)	173 (30.6)	160 (32.8)	134 (27.8)	118 (29.1)
Median [Min, Max]	168 [102, 267]	157 [62.9, 360]	132 [59.9, 223]	116 [52.8, 215]
Missing values	7 (3.3%)	15 (7.4%)	2 (0.9%)	4 (2.3%)
**Valin level**				
Low	32 (14.9%)	57 (28.2%)	124 (57.4%)	129 (75.4%)
Normal	176 (81.9%)	129 (63.9%)	90 (41.7%)	38 (22.2%)
Elevated	0 (0%)	1 (0.5%)	0 (0%)	0 (0%)
Missing values	7 (3.3%)	15 (7.4%)	2 (0.9%)	4 (2.3%)
**Alanine (umol/L)**				
Mean (SD)	361 (116)	354 (131)	337 (102)	340 (132)
Median [Min, Max]	348 [174, 862]	328 [130, 807]	316 [146, 659]	327 [124, 811]
Missing	7 (3.3%)	15 (7.4%)	2 (0.9%)	4 (2.3%)
**Alanine levels**				
Low	0 (0%)	3 (1.5%)	0 (0%)	5 (2.9%)
Normal	166 (77.2%)	144 (71.3%)	170 (78.7%)	131 (76.6%)
Elevated	42 (19.5%)	40 (19.8%)	44 (20.4%)	31 (18.1%)
Missing values	7 (3.3%)	15 (7.4%)	2 (0.9%)	4 (2.3%)
**Isoleucine (umol/L)**				
Mean (SD)	47.4 (10.2)	43.9 (12.5)	40.2 (10.1)	36.6 (10.0)
Median [Min, Max]	46.0 [28.9, 87.0]	42.1 [14.9, 161]	39.0 [18.9, 81.0]	35.6 [14.0, 74.0]
Missing values	7 (3.3%)	15 (7.4%)	2 (0.9%)	4 (2.3%)
**Isoleucine levels**				
Low	24 (11.2%)	42 (20.8%)	80 (37.0%)	94 (55.0%)
Normal	182 (84.7%)	144 (71.3%)	134 (62.0%)	73 (42.7%)
Elevated	2 (0.9%)	1 (0.5%)	0 (0%)	0 (0%)
Missing values	7 (3.3%)	15 (7.4%)	2 (0.9%)	4 (2.3%)
**Leucine (umol/L)**				
Mean (SD)	82.1 (16.5)	74.3 (17.1)	71.2 (15.8)	63.1 (17.1)
Median [Min, Max]	79.4 [49.7, 139]	72.9 [26.5, 203]	69.9 [36.6, 124]	60.5 [31.6, 137]
Missing values	7 (3.3%)	15 (7.4%)	2 (0.9%)	4 (2.3%)
**Leucine levels**				
Low	34 (15.8%)	60 (29.7%)	93 (43.1%)	113 (66.1%)
Normal	174 (80.9%)	126 (62.4%)	121 (56.0%)	54 (31.6%)
Elevated	0 (0%)	1 (0.5%)	0 (0%)	0 (0%)
Missing values	7 (3.3%)	15 (7.4%)	2 (0.9%)	4 (2.3%)
**AAT in feces (mg/g)**				
Mean (SD)	0.571 (0.603)	0.611 (0.491)	0.402 (0.335)	0.389 (0.298)
Median [Min, Max]	0.480 [0.0300, 6.36]	0.550 [0.0300, 3.52]	0.340 [0.0100, 1.80]	0.315 [0.0300, 1.48]
Missing values	11 (5.1%)	7 (3.5%)	2 (0.9%)	1 (0.6%)
**AAT levels in feces**				
Elevated	3 (1.4%)	2 (1.0%)	0 (0%)	0 (0%)
Grey zone	9 (4.2%)	11 (5.4%)	5 (2.3%)	3 (1.8%)
Missing	9 (4.2%)	4 (2.0%)	2 (0.9%)	1 (0.6%)
Normal	194 (90.2%)	185 (91.6%)	209 (96.8%)	167 (97.7%)
**Calprotectin in feces (ug/g)**				
Mean (SD)	651 (700)	888 (1280)	588 (695)	500 (580)
Median [Min, Max]	404 [55.0, 4530]	515 [54.0, 10600]	303 [56.0, 4050]	309 [72.0, 4600]
Missing values	19 (8.8%)	22 (10.9%)	17 (7.9%)	14 (8.2%)
**Calprotectin levels in feces**				
Elevated	66 (30.7%)	72 (35.6%)	56 (25.9%)	28 (16.4%)
Normal	130 (60.5%)	107 (53.0%)	142 (65.7%)	128 (74.9%)
Missing values	19 (8.8%)	23 (11.4%)	18 (8.3%)	15 (8.8%)
**CRP (mg/L)**				
Mean (SD)	4.54 (4.38)	5.16 (5.97)	6.76 (20.0)	9.36 (18.5)
Median [Min, Max]	3.00 [3.00, 31.0]	3.00 [3.00, 50.0]	0.440 [0.0100, 222]	1.11 [0.0100, 144]
Missing values	4 (1.9%)	8 (4.0%)	22 (10.2%)	23 (13.5%)
**CRP level**				
Elevated	14 (6.5%)	20 (9.9%)	30 (13.9%)	38 (22.2%)
Normal	197 (91.6%)	174 (86.1%)	164 (75.9%)	110 (64.3%)
Missing values	4 (1.9%)	8 (4.0%)	22 (10.2%)	23 (13.5%)
**Insulin-like growth factor 1 (Igf1) (ng/mL) ***				
Mean (SD)			56.5 (24.7)	54.2 (19.8)
Median [Min, Max]			56.0 [1.00, 101]	50.5 [8.00, 98.0]
Missing values			10 (4.6%)	13 (7.6%)

* Igf1 was only measured in the samples from the Central African Republic.

## Data Availability

All related, anonymized datasets can be found in the Appendix A.

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
