# Peer review of "Putative Biomarkers of Environmental Enteric Disease Fail to Correlate in a Cross-Sectional Study in Two Study Sites in Sub-Saharan Africa"

_nutrients, 2022, doi:10.3390/nu14163312_

Round 1

Reviewer 1 Report

The manuscript by Vonaesh et al. aims to evaluate the relevance of various biomarkers with EED. I have several major concerns which is as follows.

1. As gold standards for diagnosing EED were not included in this study, it is difficult to interprete the study results in relation to EED per se. It would be better to consider change the main study objectives, for example, biomarkers related to stunting (HAZ).

2. Statatistical analyes : Many variables do not seem to follow normal distribution, and needs to be corrected. Although data transformation was decribed in the method section, many figures (especially correlation analalyses) in the results section still show the data are seriously skewed.

3. Non-english (for example 'oui') are included in the mansucript.  

Reviewer 2 Report

Allow me to offer my condolences for the loss of your colleague Dr. Lova Andrianonimiadana.

This manuscript has a lot of interesting data from two settings that are under-represented in the English language scientific literature.  The key findings are interesting and important, but they get somewhat lost in the amount of data begin presented.  To this end, I have made some suggestions below.

Major comments

I think your conclusions about HAZ are misleading for two reasons:  the study is cross-sectional and the children are quite old. Currently stunted children may have had raised EED biomarkers which are responsible for their stunting months to years before their current height, even if you observe no association between EED biomarkers and HAZ in your study. The children in your study are quite old, none are under 2-years of age, and most are around 4 years of age.  The evidence suggests that the incidence of stunting is highest in much younger infants and that the prevalence of stunting peaks around 2-years of age.  It is possible most of the pathology that led to your children being stunted has been resolved.  For these reasons I do not believe your study is well place to comment on the association between the biomarkers and HAZ.  I think the biggest contributions of your work are the association between biomarkers and anemia, and the poor correlation between the biomarkers themselves.  I would tone down the HAZ inference a lot, and probably remove it from the study conclusions and abstract.  It is very misleading. 

You need to clearly state this is a cross sectional study in the methods.  And, ideally the title of the study. 

How did you adjust for selection bias were there are a lot of samples missing.  Presumably there is an underlying reason those data are missing, which could have introduced selection bias.  For biomarkers where a huge amount of data is missing – like the L:M where nearly 90% of the data is missing – I just don’t think this should be included in the paper.  For transparency you could mention in the results that it was tested, but there was too much missingness, but the data is uninterpretable with that degree of missingness.   The same applies to the duodenal Calprotectin and AAT, there is too much missingness to interpret it. 

The results are hard to follow – in part because you have lots of subsets of children with different assays.  The figure 1B is helpful, but it is hard to map this onto the text of the results sections.  As a reader it really undermines my confidence in your results that the number of children included in each section of the results is unclear.  Is there a way to group the results by the subsets of children included – I think you have a large group with most of the children, and smaller substudy of children with endocab and blood cytokines?  

It seems like some of the missingness was deliberate, for example not running certain assays in the CAR site. I would frame that decision as a sub-study in the methods.  At the moment it is frames as missing data in the results, which is confusing. 

Line 69 – you suggest micronutrient might influence the biomarkers, but it is also possible that EED (and its biomarkers) influence the micronutrients.  Ie Anemia may be caused by EED. 

From 219 onwards you start talking about the prevalence of raised biomarker values, but I can’t see a description of what cut offs you used to come up with these prevalence. Ensure that all cut-offs are defined somewhere in the methods. 

Table 2 has four columns, but I think the top header got cut off.  I think it has presenting the results by site and stunting status, but the site bit got off the top? 

How were your confounding variable selected, line 197, was this data drive or a=priori?  Clogged/runny nose is an odd one to include. 

I am not confident that methods sections describe the analysis for all results that are presented.  The HAZ analysis is not described in the methods, and the “factor associated with putative biomarkers”  also appears to be not in the methods. It needs to be easier for the reader to map the methods section onto the results, I suggest using the same sub-headers in both sections, or at least insuring they are present in the same order.  

Minor comments:

Why is the L:M not in Figure 1’s table B?

The use of “CAR” abbreviation is a bit inconsistent.  L89 it is used but without introducing what it stands for, the in Table 1 RCA is used in the column header, and L236 you use “Central African Republic”

Some issues with typos:

L73 – outcome missing a “t”. 

L74 – should read “aimed to compare”

L75 - should read ”aimed to assess”

L78 – “either of the two” instead of “any of the two”

Round 2

Reviewer 1 Report

The manuscript has been much improved after revision.

Reviewer 2 Report

Thank you for your detailed response.  It looks great now, all comments addressed.